# The Role of the Activation of the TRPV1 Receptor and of Nitric Oxide in Changes in Endothelial and Cardiac Function and Biomarker Levels in Hypertensive Rats

**DOI:** 10.3390/ijerph16193576

**Published:** 2019-09-25

**Authors:** Juan Carlos Torres-Narváez, Israel Pérez-Torres, Vicente Castrejón-Téllez, Elvira Varela-López, Víctor Hugo Oidor-Chan, Verónica Guarner-Lans, Álvaro Vargas-González, Raúl Martínez-Memije, Pedro Flores-Chávez, Etzna Zizith Cervantes-Yañez, Claudia Angélica Soto-Peredo, Gustavo Pastelín-Hernández, Leonardo del Valle-Mondragón

**Affiliations:** 1Departamento de Farmacología “Dr. Rafael Méndez Martínez”, Instituto Nacional de Cardiología “Ignacio Chávez”, 14080 Tlalpan, CDMX, Mexico; juancarlostn63@hotmail.com (J.C.T.-N.); victorhugooidor@gmail.com (V.H.O.-C.); pastelingustavo@gmail.com (G.P.-H.); 2Departamento de Patología, Instituto Nacional de Cardiología “Ignacio Chávez”, 14080 Tlalpan, CDMX, Mexico; pertorisr@yahoo.com.mx; 3Departamento de Fisiología Celular, Instituto Nacional de Cardiología “Ignacio Chávez”, 14080 Tlalpan, CDMX, Mexico; vcastrejn@yahoo.com.mx (V.C.-T.); gualanv@yahoo.com (V.G.-L.); varalv@cardiologia.org.mx (Á.V.-G.); 4Laboratorio de Cardiología Traslacional, Instituto Nacional de Cardiología “Ignacio Chávez”, 14080 Tlalpan, CDMX, Mexico; varelopz@yahoo.com; 5Departamento de Instrumentación Electromecánica, Instituto Nacional de Cardiología “Ignacio Chávez”, 14080 Tlalpan, CDMX, Mexico; raulmmemije@yahoo.com (R.M.-M.); pelfoch07@yahoo.com.mx (P.F.-C.); 6Departamento de Sistemas Biológicos, Universidad Autónoma Metropolitana Unidad Xochimilco, 04960 Coyoacán, CDMX, Mexico; zizithcervantes@hotmail.com (E.Z.C.-Y.); casoto@correo.xoc.uam.mx (C.A.S.-P.)

**Keywords:** nitric oxide, TRPV1, hypertension, capsaicin, capsazepine, *L*-NAME

## Abstract

The purpose of the present study was to analyze the actions of transient receptor potential vanilloid type 1 (TRPV1) agonist capsaicin (CS) and of its antagonist capsazepine (CZ), on cardiac function as well as endothelial biomarkers and some parameters related with nitric oxide (NO) release in *L*-N^G^-nitroarginine methyl ester (*L*-NAME)-induced hypertensive rats. NO has been implicated in the pathophysiology of systemic arterial hypertension (SAHT). We analyzed the levels of nitric oxide (NO), tetrahydrobiopterin (BH4), malondialdehyde (MDA), total antioxidant capacity (TAC), cyclic guanosin monophosphate (cGMP), phosphodiesterase-3 (PDE-3), and the expression of endothelial nitric oxide synthase (eNOS), guanosine triphosphate cyclohydrolase 1 (GTPCH-1), protein kinase B (AKT), and TRPV1 in serum and cardiac tissue of normotensive (118±3 mmHg) and hypertensive (H) rats (165 ± 4 mmHg). Cardiac mechanical performance (CMP) was calculated and NO was quantified in the coronary effluent in the Langendorff isolated heart model. In hypertensive rats capsaicin increased the levels of NO, BH4, cGMP, and TAC, and reduced PDE-3 and MDA. Expressions of eNOS, GTPCH-1, and TRPV1 were increased, while AKT was decreased. Capsazepine diminished these effects. In the hypertensive heart, CMP improved with the CS treatment. In conclusion, the activation of TRPV1 in H rats may be an alternative mechanism for the improvement of cardiac function and systemic levels of biomarkers related to the bioavailability of NO.

## 1. Introduction 

The transient receptor potential system (TRPs) participates in many biological functions in different species, including humans. The cardiovascular function of type 1 and 2 subfamilies of vanilloid TRPs (TRPV1, TRPV2) has been well defined in physiological studies [1]. However, its potential role in physiopathological conditions such as systemic arterial hypertension (SAHT) requires more experimental and clinical studies [2,3,4,5,6]. Randhawa et al. have linked TRPV1 activation on the vascular endothelium to the vasodilator actions of nitric oxide (NO) and also to the neuropeptide calcitonin gene related-peptide (CGRP). However the mechanisms of action are still not well known [7].

The TRPV1 is located in different cell types in mammals such as nerve endings [4,5,7] and endothelial cells of several vascular zones. The TRPV1 responds to mechanical, thermal, and chemical stimuli due to its permeability to ions such as Na^+^, Ca^2+^, and K^+^ [8,9,10,11,12,13,14]. The TRPV1 participates in the control of vascular smooth muscle tone and consequently, on regulation of arterial pressure [6,7,15,16]. Its endogenous activation is exerted by the endocannabinoid, anandamide [4,12,16,17,18]. Moreover, it has been demonstrated that the binding site for capsaicin (an agonist) on the TRPV1 (S512), is the same as that of the endogenous agonist anandamide. In these agonist molecules, the amide group is the receptor binding region and the effects of both are antagonized by capsazepine [18].

In some pathological conditions, such as in SAHT, NO synthesis is impaired because there is an enzymatic uncoupling of endothelial nitric oxide synthase (eNOS) due to either tetrahydrobiopterin (BH4) deficiency or to guanosine triphosphate cyclohydrolase 1 (GTPCH-1) deficiency. GTPCH-1catalyzes BH4 formation through guanosine triphosphate (GTP) hydrolysis and regulates the levels of BH4 in tissues and blood; thus, the production of the superoxide anion (∙O_2_^−^) increases [7,19,20,21].

The reduction in NO production and the increase in reactive oxygen species (ROS) diminish the bioavailability of NO in SAHT. The relationship of NO with its biological targets including cardioprotective biomarkers is also decreased [22,23,24]. These events produce a generalized diminution in the total antioxidant capacity (TAC) [25]. These alterations cause an increase in lipoperoxidation of the cellular membranes, thus increasing the plasmatic levels of malondialdehyde (MDA) [26].

Another example of this type of damage on biological mechanisms occurs in vascular smooth muscle cells, where NO activates soluble guanylate cyclase (sGC) [19] to transform GTP into cyclic guanosine monophosphate (cGMP). In pathological states, cGMP is degraded to guanosine monophosphate (GMP) by the action of phosphodiesterases, particularly phosphodiesterase-3 (PDE-3), thereby inhibiting the relaxing effect in vascular smooth muscle [27].

The aim of this work was to explore the participation of the TRPV1 receptor in NO release at the systemic and cardiac levels, as well as to analyze the effects of activating and/or inhibiting the TRPV1 on mechanical activity of the heart and on some biomarkers related to NO production and oxidative stress, including eNOS, TAC, dihydrobiopterin (BH2), BH4, MDA, PDE-3, and cGMP. For this purpose, we used a SAHT model in rats which is induced with L-nitro-arginine methyl ester (L-NAME).

## 2. Material and Methods

### 2.1. Reagents and Antibodies

Capsaicin [8-methyl-N-vanillyl-6-nonenamide]; capsazepine [N-(2-(4-chlorophenyl)ethyl)-1,3,4,5-tetrahydro-7,8-dihydroxy-2H-2-benzazepine-2-carbothioamide]; *L*-nitro-arginine methyl ester, and all reagents used in this study were from Sigma Chemical Co., St. Louis, MI, USA.

The following antibodies were used: eNOS (H-159) rabbit polyclonal IgG, β-actin (H-196) rabbit polyclonal IgG, goat-anti-rabbit IgG-HRP conjugated, and TRPV1 (VR11-A) rabbit anti-rat VR1 antibody (Santa Cruz Biotechnologies, Santa Cruz, CA, USA).

### 2.2. Animals

Experimental animals (male Wistar rats 300–350 g body weight) were obtained from the animal facility of the National Institute of Cardiology Ignacio Chávez in Mexico City, after approval by the Institutional Ethics Committee on the use and care of experimental animals.

Animals were kept under normal light conditions (12 h light/darkness), at a controlled temperature (25 ± 3 °C) and humidity (50 ± 10 %); animals were fed ad libitum with a standard diet (certified diet for rat, LabDiet 5026, PMI Nutrition International, Richmond, IN, USA) and water. All procedures followed the guidelines established by the Federal Regulation for Experimentation and Animal Care (SAGARPA, NOM-062-ZOO-1999, México) [28].

### 2.3. Experimental Groups

Animals were divided into 8 groups with 6 animals each, as follows: (1) normotensive (N), (2) N + CS, (3) N + CZ, (4) N + CZ + CS, (5) hypertensive (H), (6) H + CS, (7) H + CZ, (8) H + CZ + CS. All groups had free access to water and food. SAHT of the rats (groups 5 to 8) was induced by addition of *L*-NAME (200 mg/L) to the drinking water during 40 days [29,30,31]. On day 36 of *L*-NAME treatment, the following treatments were applied during 4 days before the experiment: Group 2 and 6 received a subcutaneous dose (0.3 mL) of CS (5 mg/Kg weight, dissolved in ethanol-water (2:1). The administration was done at 10:00 a.m.) [31]. Group 3 and 7 received a subcutaneous (0.3 mL) dose of CZ (4 mg/Kg weight, dissolved in ethanol-water (2:1), for 4 days. The administration was done at 10:00 a.m.) [16,32]. A combination of CZ+CS was applied to the groups 4 and 8. CZ was applied first (0.3 mL s.c.), and CS (0.3 mL s.c.) was applied one hour later. All administrations of CS, CZ, and CS + CZ were made at 10:00 a.m. *L*-NAME treatment did not stop during the treatments and the animals were euthanized on day 40.

Mean arterial pressure (MAP) was measured in all groups by a non-invasive method, using a pneumatic pressure monitor placed on the base of the animal’s tail, at the start and end of the treatments [33]. CS and CZ were diluted in ethanol–water at a 2:1 proportion.

One group of animals was used to obtain serum and cardiac tissue samples to perform the different analyzes. Another group of animals from all of the experimental groups was used to extract and perfuse the heart according to the Langendorff’s method and to study the cardiac mechanical activity.

### 2.4. Determinations in Blood and Ventricular Tissue

For NO, TAC, BH4, BH2, and MDA determinations, blood and left ventricular tissue samples were obtained. Serum was separated by centrifugation at 2500 rpm for 5 min at 10 °C (Sorvall SR70, Thermo Scientific Inc., Urbana, IL, USA), and stored at −70 °C until analyzed. On the day of analysis, samples were gradually thawed and deproteinized at a 10:1 proportion with 20% trichloroacetic acid (v:v), then centrifuged at 16,000× *g* during 15 min at 10 °C (Sorvall SR70, Thermo Scientific Inc., Urbana, IL, USA). Supernatants were filtered through a 0.22 μm nitrocellulose filter (Millipore, Billerica, MA, USA). Tissue (small sections of the left ventricle) was obtained from all euthanized animals. It was immediately frozen in liquid nitrogen and stored at −70 °C until used. On the day of analysis, samples were homogenized in cold, 5 mM phosphate buffer at pH 7.4; then, centrifuged at 16,000× *g* during 15 min at 10 °C (Sorvall SR70, Thermo Scientific Inc., Urbana, IL, USA). Supernatants were filtered through a 0.22 μm nitrocellulose filter (Millipore, Billerica, MA, USA).

The determinations of NO and TAC were performed directly on the filtered supernatant, whereas for BH4, BH2, and MDA determinations, supernatants were diluted 1:10 with 0.1 M NaOH before being analyzed.

In our study, the cGMP was determined only in the ventricular tissue. On the day of analysis, ventricular fragments were homogenized independently in a cold 100 mM phosphate buffer at pH 7.5 ± 0.05. Then, homogenates were centrifuged at 16,000× *g* during 15 min at 10 °C (Sorvall SR70, Thermo Scientific Inc., Urbana, IL, USA). Supernatants were filtered through 0.22 μm nitrocellulose filters (Millipore Billerica, MA, USA); diluted 1:10 with 0.05M NaOH and analyzed directly at 10 °C.

PDE-3 levels were measured in left ventricular tissue [34]. For this, the fresh tissue was rapidly homogenized in a cold buffer of 100 mM saccharose + 20 mM HEPES + 50 mM citrates at pH 5.6. Then, homogenates were centrifuged at 16,000× *g* during 15 min at 10 °C (Sorvall SR70 centrifuge, Thermo Scientific Inc., Urbana, IL, USA). Supernatants were filtered through 0.22 μm nitrocellulose filters (Millipore Billerica, MA, USA); then diluted 1:10 with 0.1 M NaOH and filtered again with Sep-Pak Classic C18 (Waters, Urbana, IL, USA) filters to be directly analyzed [35].

All samples (liquid and tissue) were kept at −70 °C until the beginning of the respective assay.

### 2.5. Measurement of Nitric Oxide

Quantification of NO was carried out in the sera of rats and filtered supernatants (0.22-μm nitrocellulose filter, Millipore, Billerica, MA, USA) of ventricular tissue homogenate [36]. In another group of animals with the different treatments, the heart was isolated and connected to a Langendorff’s apparatus and samples of coronary effluent were obtained to measure NO. All determinations were carried out using a DW2000 spectrophotometer (SLM-Aminco, SLM Instruments Inc., Urbana, IL, USA) in UV-Vis region (490 nm), at room temperature.

### 2.6. Measurement of Total Antioxidant Capacity

TAC was quantified in sera of rats [37], by a spectrophotometric method in UV-Vis region, at room temperature for which a UV-visible spectrophotometer (Cary 4000, Varian Inc., Mulgrave, Victoria, Australia) at 490 nm.

### 2.7. Measurement by Capillary Zone Electrophoresis

The capillary zone electrophoresis (CZE) technique was used to quantify the biomolecules BH4, BH2, MDA, cGMP, and PDE-3 in sera and/or ventricular tissue of rats. The analysis was carried out using the system P/ACETM MDQ (Beckman Coulter Inc., Fullerton, CA, USA), with UV-Vis detection by diode array. In sera, the sample was deproteinized with 1:10 methanol, centrifuged at 15,000× *g* for 15 min at 10 °C (Sorvall SR70, Thermo Scientific Inc., Urbana, IL, USA) and filtered through nitrocellulose membrane 0.22 µm (Millipore, Billerica, MA, USA), then it was diluted 1:1 with 0.1 M sodium hydroxide. In ventricular tissue, the filtrate of homogenate was deproteinized with cold methanol and then with cold 10% trichloroacetic acid, both in 10:1 ratio. It was centrifuged at 16,000× *g* for 15 min at 10 °C (Sorvall SR70, Thermo Scientific Inc., Urbana, IL, USA) and filtered through nitrocellulose membrane of 0.22 µm (Millipore, Billerica, MA, USA), and then it was diluted 1:10 with cold 0.1 M sodium hydroxide. The sample was passed through a cold Sep-Pak Classic C-18 cartridge (Waters, Urbana, IL, USA) pretreated with 10 mL of a 100 mM citrate buffer pH 2.5.

BH4 and BH2 were determined using the methodology of Han et al. [38]. The analysis was performed to 30 kV during 6 min at a wavelength of 230 nm at 10 °C using the running buffer (0.1 M Tris—0.1 M boric acid—2mM EDTA, pH 8.75). The sample was injected under hydrodynamic pressure to 0.5 psi/10s.

MDA was determined using the methodology of Cleason et al. [25]. The analysis was performed at −20 kV during 4 min, at a wavelength of 267 nm at 10 °C using 100 mM borates + 0.5 mM CTAB at pH 9.0 buffer. The sample was injected under hydrodynamic pressure to 0.5 psi/10s.

cGMP was determined according to method of Friedecky et al. [39]. The analysis was performed at −25 kV for 15 min at 190 nm using the running buffer (40 mM citric acid + 0.8 mM CTAB at pH 4.4). The sample was injected under hydrodynamic pressure to 0.8 psi/10s.

PDE-3 was determined according to the methodology of Yan et al. [27]. The analysis was performed at 20 kV for 30 min at 240 nm at 20 °C and the running buffer (100 mM boric acid at pH 2.8). The sample was injected under hydrodynamic pressure to 0.7 psi/10s.

### 2.8. Isolated Heart Perfused by the Langendorff’s Method

Animals of all experimental groups were anesthetized with sodium pentobarbital (60 mg/Kg body weight) and anticoagulated with heparin (1000 U/mL/Kg body weight). The heart was exposed through a thoracotomy and the ascending aorta was referred with a silk thread. The heart was then quickly removed, placed in isotonic saline at 4 °C, and immediately connected to the perfusion system through the ascending aorta [40]. Once the heart was connected, it was given an adaptation period of 30 min to the new perfusion conditions: 5 min with a flow (F) of 25 mL/min and 25 min with a F of 12 mL/min. Heart rate (HR) was maintained at 312–324 beats per minute, by the use of a Grass stimulator (U7, Grass Instruments Co., Quincy, MA, USA). Coronary flow was held at 12 mL/min with a peristaltic pump (SAD22, Grass Instruments Co., Quincy, Mass., USA) throughout the experimental period. After the adaptation time, liquid effluent samples were taken every 10 min. During all of the experiment, parameters such as left intraventricular pressure (LIVP) were recorded by using a latex balloon attached to a Grass hydropneumatic pressure transducer. The balloon was placed by advancing a catheter through the mitral valve into the left ventricle and once inside of the cavity, an internal pressure of 5–10 mmHg (diastolic pressure) was applied. Perfusion pressure (PP) was also recorded and a range of 55–70 mmHg at the start of the experiment was considered as an inclusion criterion. Cardiac mechanical performance (CMP) was calculated as HR x LIVP = CMP. Coronary vascular resistance (CVR) was calculated using the following relationship: PP/F = RVC [41].

### 2.9. Western Blotting of GTPCH-1, eNOS, AKT, and TRPV1

eNOS, GTPCH-1, protein kinase B (AKT), and TRPV1 were analyzed in ventricular tissue. Protein extraction was carried out using a lysis buffer (1 M Tris pH 7.4, 2.5 M NaCl, 0.2 M EDTA, 1% Triton X-100, 0.1 M 2-mercaptoethanol, 1% deoxycholate) supplemented with a mixture of protease inhibitors (leupeptin, aprotinin, pepstatin, PMSF) (Sigma Chemical Co., St. Louis, Missouri, USA). Tissue was homogenized at 4 °C in the lysis buffer, using a disperser (DIAX 900, Heidolph Instruments, Schwabach, Germany). Protein was determined by Bradford assay (protein assay, Bio-Rad Laboratories) [42].

A total of 50 μg of protein from the ventricular tissue homogenate was mixed 3x with loading buffer (30% glycerol, l.6% SDS, 3% bromophenol blue, 5% 2-mercaptoethanol, 125 mM Tris, pH 6.8), separated by SDS-PAGE (10% bis-acrilamide-Laemmli gel), and transferred to a 0.22-μm polyvinylidene difluoride (PVDF) membrane. Blots were blocked for 1 h at room temperature with Tris-buffered saline plus 0.01% Tween-20 (TBS-T) and 5% non-fat milk. The membranes obtained were incubated with primary antibodies anti- TRPV1 (V2764) (from Sigma-Aldrich, St. Louis, MO, USA), anti-NOS3 (sc-376751) (Santa Cruz Biotechnology, Santa Cruz, CA, USA), anti-AKT and anti-GTPCH-1 (Calbiochem, La Jolla, CA, USA) overnight at 4 °C. The membranes were incubated overnight at 4 °C with horseradish peroxidase conjugated secondary antibodies, dilution 1:10,000 (Santa Cruz Biotechnology). Blots incubated with anti-NOS3 (sc-376751), anti-AKT (sc-135829), and anti-GTPCH-1 (sc-69962) were then incubated with anti-β-Actin (sc-81178) (Santa Cruz Biotechnology, Santa Cruz, CA, USA) as control; while, blots incubated with TRPV1 were incubated with anti-GAPDH (sc-365062) (Santa Cruz Biotechnology, Santa Cruz, CA, USA) as control. Protein was detected by chemiluminescence assay (Clarity Western ECL Substrate, Bio-Rad Laboratories, Inc., Hercules, CA, USA). Chemiluminescence emitted in this process was detected in X-ray films (AGFA, Ortho CP-GU, Agfa HealthCare NV, Mortsel, Belgium). Images from each film were acquired with a GS-800 densitometer (including Quantity One software from Bio-Rad Laboratories, Inc.). The values of each band density are expressed as arbitrary units (AU).

### 2.10. Statistical Analysis

Statistical analysis was performed with software Grap Pad for Windows (SPSS Inc., Chicago, IL, USA), using a Newman’s post hoc test. All results are expressed as mean ± standard deviation and a significance of *p* ≤ 0.05 was considered.

## 3. Results

A significant difference in the mean arterial pressure (MAP) was found when normotensive rats were compared to *L*-NAME-treated rats (Table 1). When a comparison of H against H plus CS was done, it was observed that the TRPV1 activation attenuated the MAP in hypertensive rats. CZ treatment induced an increase of MAP in the hypertensive rats and CZ plus CS treatment attenuated it with a slight decrease in MAP in hypertensive rats in comparison with their own controls (Table 1).

### 3.1. Cardioprotective Biomarker Concentrations in Serum and in Tissue. Differences between Control and Hypertensive Rats

There was a reduction (of at least in 60%) in serum levels of cellular protective biomarkers NO, BH4, TAC (Figure 1A,B,D, respectively) and in cGMP in the ventricular tissue of hypertensive rats (Figure 1C). The same parameters were also compared in normotensive rats with the different treatments. In contrast, biomarkers related to cellular stress and damage such as PDE-3 and MDA increased 3 and 8 times respectively (Figure 1E,F).

### 3.2. Effects of CS and CZ on Cardioprotective Biomarkers in Sera of N and H Rats

NO serum concentration increased 66% in the N rats under the action of CS, and 3.5 times in H rats. The increase of NO concentration in the H rats reached the level of the N control rats. CZ reduced (60%) NO in N rats and it increased it 2.5 times in H rats. Conjoint administration of CZ and CS increased the serum levels of NO reaching almost the level found in control N rats (Figure 2A).

BH4 levels increased after CS administration in both N as in H rats; particularly in the last ones, where the increase was more significant. CZ and CS administered alone or in combination rose BH4 levels (Figure 2B).

Capsaicin administered to normotensive rats produced a 1.7 fold increase in TAC. In the hypertensive rats whose TAC was reduced by 60% when compared to that of normotensive animals, there was a restitution of the level of this parameter reaching almost the levels of activity found in the control normotensive rats. Administration of CZ alone or associated to CS, did not modify the levels of TAC in N and H rats (Figure 2C).

Serum concentration of MDA, the biomarker of lipid peroxidation in sarcolemma, was 8 to 10 times higher in H rats when compared to that in N rats, and it was reduced in the hypertensive rats after administration of CS, CZ, and CS plus CZ. Although this effect was statistically significant, it did not reach the levels found in the non-hypertensive rats (Figure 2D).

### 3.3. Cardiac Mechanical Performance in Normotensive and Hypertensive Rats

Figure 3A shows the CMP of isolated hearts of N and H rats, with the different treatments. CS did not alter the CMP of normotensive hearts but CZ alone and in combination with CS produced a significant decrease in CMP. In H hearts, CS produced an improvement of CMP which was diminished by CZ and CZ plus CS treatments. When we compared N vs. H, CMP was diminished by 25% in H hearts.

### 3.4. Nitric Oxide Levels in Coronary Effluent of Normotensive and Hypertensive Hearts

NO levels in the coronary effluent (Figure 3B) of N and H isolated hearts showed important differences (40 ± 1 and 14 ± 4 pmoles/mL respectively). NO levels were significantly decreased in the H hearts. CS produced an increase in the NO release in N and H conditions by 36 % and 46 % respectively when compared with their own controls. CZ and CZ plus CS treatments produced a decrease in NO levels. The NO decrease was of 82 % by CZ and of 73 % by CZ plus CS in N hearts. NO was decreased by 85 % by CZ and by 71 % by CZ plus CS in H hearts when compared with N plus CS and H plus CS treatments respectively.

### 3.5. Coronary Vascular Resistance in Normotensive and Hypertensive Hearts

Table 2 shows the significant differences observed when comparing the CVR in hearts of normotensive and hypertensive rat hearts (4 to 11 mmHg/mL/min on average respectively). Importantly, CS and CZ + CS treatments induced a significant decrease on CVR of hearts obtained of hypertensive rats when compared to those obtained from the H group.

### 3.6. BH2 and BH4 Measured in Cardiac Tissue

We measured BH4 and BH2 levels in the ventricular tissue to determine their effect on NO production in normotensive and hypertensive rats. The basal BH4 levels (0.0988 ± 0.0081 pmol/mg of tissue) were significantly increased (to 0.1792 ± 0.017 pmol/mg of tissue) by CS treatment and significantly decreased by CZ treatment (to 0.064 ± 0.0048 pmol/mg of tissue) in the normotensive conditions (Figure 4A). The BH4 levels were significantly diminished in the H group (0.039 ± 0.0018) when compared to the N group (0.10 ± 0.003 pmol/mg of tissue). The activation of TRPV1 receptor with CS in H group stimulated significantly the BH4 production (0.1595 ± 0.007 to 11.1 ± 0.2). CZ also stimulated BH4 levels to 0.1089 ± 0.0057. BH4 levels reached 0.2363 ± 0.0129 pmol/mg of tissue with CZ plus CS.

The BH2 levels (Figure 4B) did not show significant differences in the N group against the N plus the treatments groups. In the hypertensive conditions, BH2 levels were increased 4 fold when compared with the N group. This significant increase was prevented by the activation of TRPV1 with CS but also with CZ. This effect was also observed with CZ plus CS treatment.

### 3.7. NO, cGMP, PDE-3 in Ventricular Tissue

The levels of NO, cGMP, and PDE-3 in ventricular tissue were measured (Figure 4) and a correlation between their levels in normotensive conditions and the pathological state was found. NO release (Figure 4C) and production of cGMP (Figure 4D) were increased in hearts from normotensive rats treated with CS when compared to their respective controls. CS stimulated cGMP formation by 100% in normotensive conditions and this effect was decreased by the CZ action. It is noteworthy that in hypertension, the levels of NO and cGMP were diminished even when the treatments were applied. Although NO levels in tissue recovered to baseline levels, cGMP remained at low levels even with treatment.

PDE-3 levels (Figure 4E) significantly decreased with the three different treatments in normotensive conditions. PDE-3 levels were elevated in the hearts of hypertensive rats when compared to the N group. CS and CZ plus CS maintained the variables at the baseline levels.

### 3.8. GTPCH-1 Expression

Figure 5 shows that hypertension decreased GTPCH-1 expression (Panel A) when compared to its expression in N rats. The treatment with CS increases GTPCH-1 expression in H rats when compared to its expression in H rats without treatment. In N and H rats treated with CZ, there were no changes in GTPCH-1 expression when compared with controls. Interestingly in H rats, CS + CZ treatment increased GTPCH-1 expression when compared to that in H + CS rats.

### 3.9. eNOS Expression

Our data show that eNOS expression (Figure 5B) decreased in the hypertensive rats when compared to that in N rats. Treatment with CS increased eNOS expression in H rats when compared to that in control rats. CZ did not generate changes in N and H conditions compared to their respective controls. The combination of treatments produced an increase in the eNOS expression in the H group.

### 3.10. AKT Expression

There was a decrease in AKT expression in the ventricular tissue from tissue taken from hypertensive rats when compared to tissue taken from rats in the N condition (Figure 5C). CS, CZ, and the combination produced a decrease in AKT expression when comparing it to the respective controls (N and H rats).

### 3.11. TRPV1 Expression

TRPV1 expression in ventricular tissue of the N + CS group was increased when compared to the N group without treatment (Figure 6). CZ and CZ plus CS did not affect the TRPV1 expression in the N group. In the cardiac tissue of hypertensive rats there were no significant differences in TRPV1 expression with respect to the N group.

## 4. Discussion

In this paper we explored the participation of the TRPV1 receptor in NO release at the systemic and heart levels, and analyzed the effects of activating and/or inhibiting the TRPV1 on the mechanical activity of the heart and on some biomarkers related to NO production and oxidative stress, such as: eNOS, TAC, BH2, BH4, MDA, Phosphodiesterase-3 (PDE-3), and cGMP. A systemic arterial hypertension model in rats which is induced with L-nitro-arginine methyl ester (L-NAME) was used. The comparative analysis of NO, BH4, cGMP, TAC, PDE-3, MDA concentration levels from N and H rats is consistent with the genesis of a hypertensive state induced by the administration of *L*-NAME. Our results suggest that the hypertensive state reduces the cardiovascular protective pathways (NO, BH4, cGMP, TAC) and increases the expression of damage biomarkers such as MDA and of cGMP degradation inductors such as PDE-3. Through these actions, the hypertensive state might promote ROS formation whose increase is associated to a NO elevation that results from an imbalance in the endogenous antioxidant system. These effects promote instability in the molecular regulatory processes [42,43,44,45,46,47,48].

The results of the present paper corroborate that the TRPV1 receptor plays an important role in the regulatory mechanisms of arterial pressure, particularly in experimentally induced arterial hypertension in rats. A differential susceptibility between N and H rats to the actions of CS and of CZ on eNOS expression was found. If *L*-NAME inhibits NOS action, why can NO be generated by TRPV1 activation? A possible explanation of this fact could be that inhibition of eNOS, by *L*-NAME is not total. This is probably due to the fact that the *L*-NAME molecule, has not an active blocking effect by itself; *L*-NAME has to be previously hydrolyzed by the action of esterases to free *L*-NOARG (*N^G^*-nitro-*L*-free arginine) to be able to inhibit actively NOS [42]. The polyphenolic chemical nature of CS and of CZ results in inhibitory actions on a large variety of esterases [43,44]. This characteristic of CS and CZ can be responsible for the NO viability in hypertensive animals. This could be favored by Ca^2+^ entering through active TRPV1. Additionally, the phenolic groups of CS and of CZ act as scavengers for free radicals [45] and counteract the ROS-produced damage. CS or CZ or their combination (CZ plus CS) in N and H rats diminished the endothelial damage caused by oxidative stress considering the reduction of serum levels of MDA a biochemical marker of cell lesion.

Our results confirm that BH4 was diminished or became inactive in the *L*-NAME hypertension model (Figure 1B), which correlates with the diminution in NO levels in the sera of hypertensive rats (Figure 1A). BH4 is an essential factor for the three NOS isoforms [20,23] and its level in the endothelium is critical for the regulation of NO and ROS production; hence, this cofactor as an indicator of possible alterations in NO production and, consequently, of alterations in systemic arterial pressure [22].

CS treatment generated an overproduction of BH4 in H rats (Figure 2B), probably because the damaged endothelium was unable to regulate the bioavailability of this cofactor and of NO. ROS production increased and the total antioxidant capacity diminished at the systemic level in the hypertensive conditions [25,46]. The antagonism produced by CZ strengthens the idea that TRPV1 importantly participates in NO release at the systemic level.

CS and CZ are molecules with phenolic groups that act as scavengers of free radicals, as well as inactivators of the superoxide anion [45]. Due to their antioxidant activity, they contribute to diminish the damage produced by ROS. MDA is an indirect indicator of toxic levels of the superoxide anion that damages cellular membranes. Our results showed that treatments with CS or CZ or their combination (CZ plus CS) in N and H rats diminished the damage caused by the oxidative stress (Figure 2D). This effect becomes more evident in the pathological status.

TAC deficiency indicates that the endogenous antioxidant system is unstable in hypertensive rats (Figure 2C) and, in consequence, ROS is increased, inducing an altered NO production (Figure 2A), which, in turn, generates endothelial dysfunction.

The phenolic characteristics of CS and CZ also contribute to the inhibition of phosphodiesterases [43,44], favoring vasodilation through the nitric oxide/soluble guanylate cyclase (NO/sGS) pathway. CS avoids degradation of cGMP [47,48,49], through at least two mechanisms: a) an increase in intracellular Ca^2+^ concentration to activate eNOS, which can be blocked by *L*-NAME, b) phosphorylation of eNOS through the PI3-kinase/Akt pathway, which cannot be blocked by *L*-NAME [50]. Actually, this topic is being investigated in our laboratory.

Our results confirmed that activation of TRPV1 induces a decrease of phosphodiesterase-3 in cardiac tissue (Figure 2E).

Biochemically, our results of TRPV1 activation with CS correlate with an improvement in the MAP of hypertensive rats prior to experiments and a possible explanation is an increase of cGMP (Figure 2C), a decrease in PDE-3 and an increase of NO bioavailability.

The CS treatment is related to an improvement in the CMP (Figure 3A) and CVR (Table 2) with an increase on NO levels (Figure 3B) in the isolated hearts perfused through the coronary arteries. There is an evident improvement on CMP and CVR which correlates with increases in the BH4, NO, and cGMP levels in the ventricular tissue of rats receiving the CS treatment.

On the other side, the expression of eNOS in the cardiac tissue of N vs. H rats (Figure 5B) showed important differences, which could explain in part the diminution of the NO levels quantified in sera of H rats.

There were differences in the eNOS expression when comparing N against H rats. However, there were no significant differences in eNOS expression by the action of CS or CZ in the hearts of N rat. In H hearts the drugs (CS and CZ) stimulated eNOS and GTPCH-1 expression preventing the development of hypertension.

As is known, eNOS activation is a complex process that depends on several factors such as BH4 levels and Ca^2+^ concentration. Furthermore, this isoenzyme can be phosphorylated by different proteins such as AMPK, Akt, PKC in serine or threonine regions. In addition, there are other mechanisms that regulate its activation such as acylation, nitrosylation, glycosylation [51].

Based on our present results, we can state that treatment with capsaicin diminishes hypertension by restoring the homeostasis of damage biomarkers and of some parameters related with NO synthesis. An improvement in the physiological functioning of some components of sGC pathway was generated with TRPV1 activation.

We propose that the TRPV1 receptor is a cellular structure that plays a relevant role in the regulation of factors directly involved in the mechanisms of NO synthesis in N and H rats at the systemic level and directly in the ventricular tissue [52].

NO is even increased in hypertensive animals and this might be due to the presence of pathologic stress that results in an increase in ROS [22]. CZ may trap and inhibit ROS, decreasing oxidative stress and it may activate compensatory NO releasing mechanisms through its molecular nature as a polyphenol (free radical scavenger) [53]. This paradoxical effect of CZ is similar to the effect of the agonist CS. Thus, the effects of the agonists and antagonists of TRPV1 are different in the normotensive and hypertensive states.

Paradoxical and unexpected results are often observed when comparing the effects of CS and CZ in control and hypertensive organisms as illustrated by their effects on MDA (Figure 2D) and BH2 levels (Figure 4B). These variables showed a similar decrease in hypertensive rats treated with CZ and in the hypertensive group without treatment. In this group no antagonic effect of CZ is observed. This could be the consequence of the reductive effect of CZ which leads to a decrease in ROS and a diminution of damage to the cellular membrane by decreased oxidative stress.

## 5. Conclusions

This study corroborates that TRPV1 activation stimulates NO production and is also involved in BH4 and cGMP regulation. Furthermore it significantly increased the total antioxidant capacity. The study also demonstrates that CS diminished MDA production, suggesting a reduction in the damage induced by cell membrane lipoperoxidation when cells are exposed to the oxidative stress generated during hypertension. The CS-induced reduction in PDE-3 levels, protects cGMP from degradation, fostering the signaling of second messengers involved in vasodilation and reducing the damage caused by oxidative stress. Finally, TRPV1 stimulation by CS significantly participates in a cardiovascular protection effect during SAHT.

## Figures and Tables

**Figure 1 ijerph-16-03576-f001:**
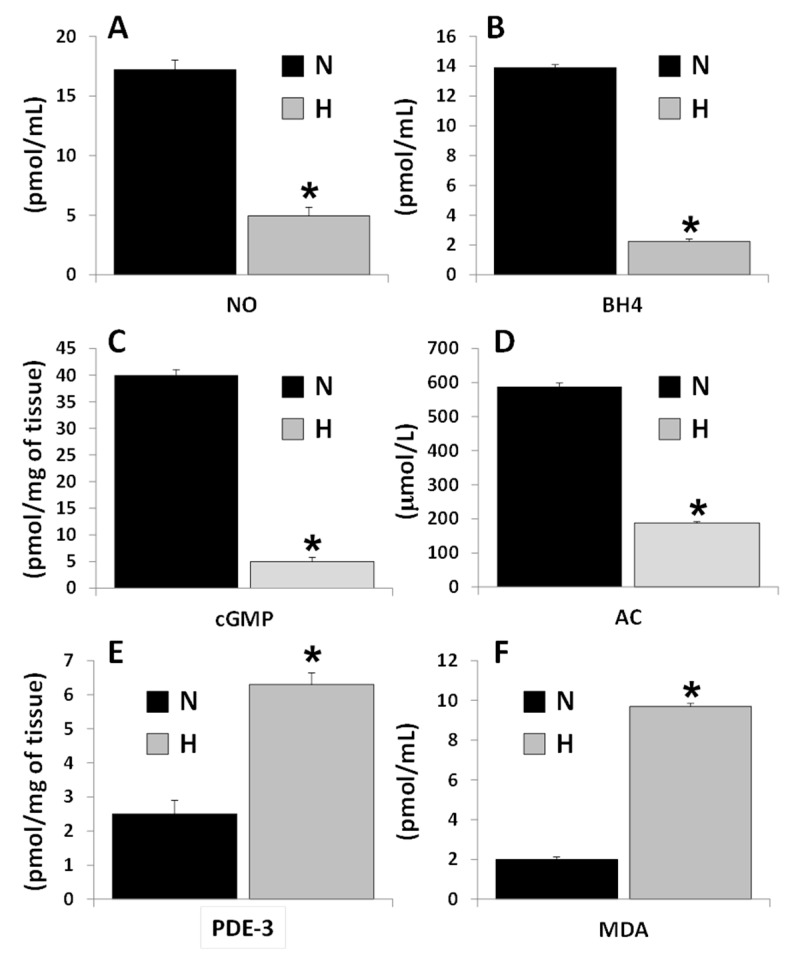
Comparison of the levels of nitric oxide (NO) (**A**), tetrahydrobiopterin (BH4) (**B**), total antioxidant capacity (TAC) (**D**), malondialdehyde (MDA) (**F**), in the sera and cyclic guanosin monophosphate (cGMP) (**C**), phosphodiesterase-3 (PDE-3) (**E**) in the ventricular tissue of normotensive (N) and hypertensive (H) rats. Values represent the mean ± standard deviation of *n* = 6, * *p ≤* 0.05, by Student’s *t*-test.

**Figure 2 ijerph-16-03576-f002:**
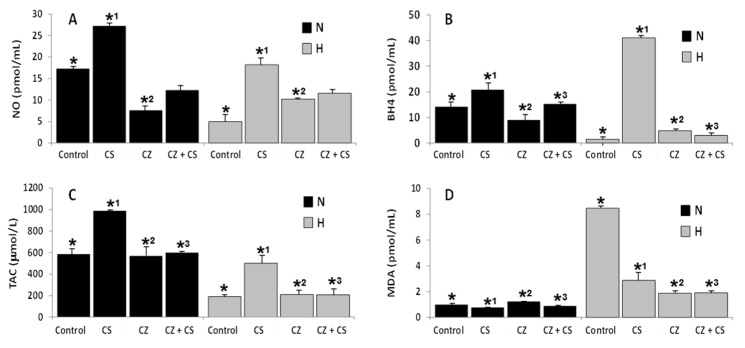
Effects of the capsaicin (CS) and capsazepine (CZ) on levels of nitric oxide (NO) (**A**), tetrahydrobiopterin (BH4) (**B**), total antioxidant capacity (TAC) (**C**), and malondialdehyde (MDA) (**D**), in the sera of normotensive (N) and hypertensive (H) rats. Values represent the mean ± standard deviation of *n* = 6, (*, *^1^, *^2^, *^3^) *p ≤* 0.05, by Newman’s post hoc test.

**Figure 3 ijerph-16-03576-f003:**
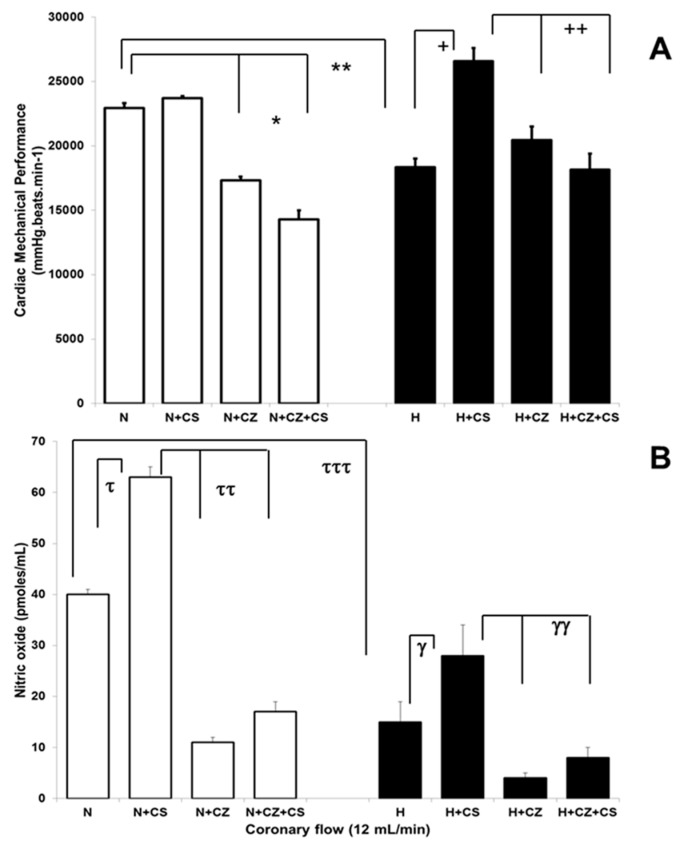
Cardiac mechanical performance (**A**) and nitric oxide levels in liquid effluent (**B**) in isolated heart of normotensive and hypertensive rats (withe and black bars respectively) in presence or absence of capsaicin (CS) and capsazepine (CZ) treatments. Values represent the mean ± standard deviation of *n* = 6, (*, **, ^+^, ^++^, ^τ^, ^ττ^, ^τττ^, ^γ^, ^γγ^) *p ≤* 0.05, by Newman’s post hoc test.

**Figure 4 ijerph-16-03576-f004:**
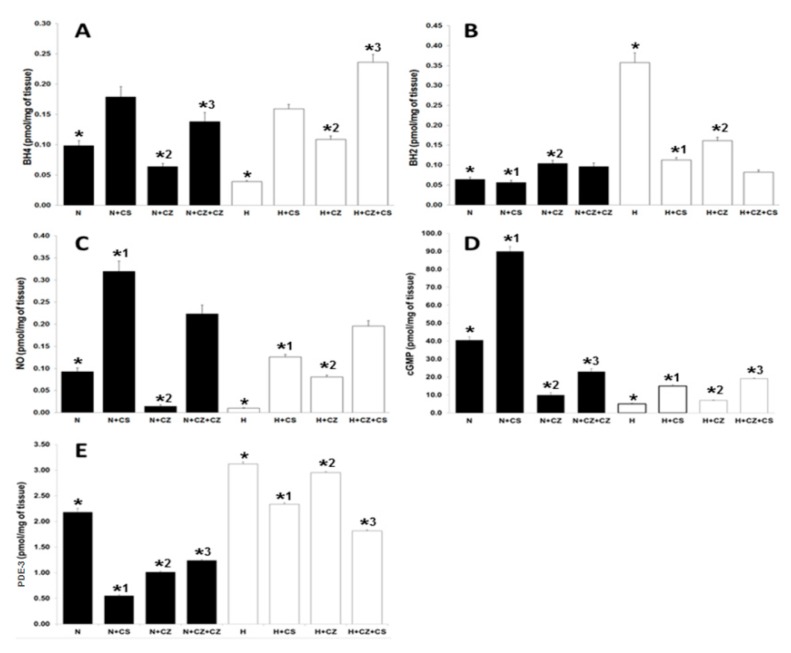
Evaluation of the levels of the following parameters in presence or absence of the different treatments. Tetrahydrobiopterin (BH4) (panel **A**), dihydrobiopterin (BH2) (panel **B**), nitric oxide (NO) (panel **C**), cyclic guanosine monophosphate (cGMP) (panel **D**), phosphodiesterase-3 (PDE-3) (panel **E**), in ventricular tissue obtained from normotensive (black bars) and hypertensive (white bars) rats. Values represent the mean ± standard deviation of *n* = 6, (*, *^1^, *^2^, *^3^) *p* ≤ 0.05, by Newman’s post hoc test.

**Figure 5 ijerph-16-03576-f005:**
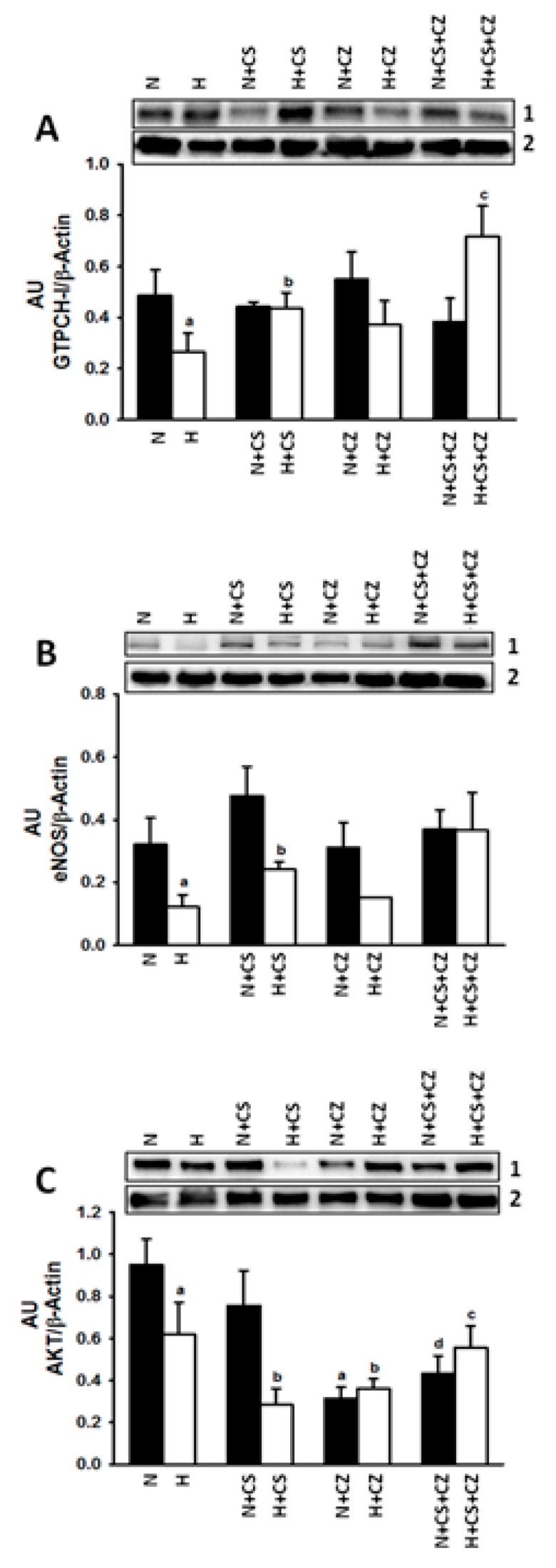
Expression of guanosine triphosphate cyclohydrolase 1 (GTPCH-1), endothelial nitric oxide synthase (eNOS), protein kinase B (AKT) in ventricular tissue from normotensive and hypertensive rats (black and white bars respectively) with the different treatments. Panel (**A**): GTPCH-1 expression in ventricular tissue (*n* = 3–4 rats/group); a = *p* < 0.05 vs. Normotensive (N), b = *p* < 0.05 vs. Hypertensive (H), c = *p* < 0.05 vs. Hypertensive + Capsaicin (H + CS). Panel (**B**): eNOS expression in left ventricle (*n* = 3–4 rats/group); a = *p* < 0.05 vs. Normotensive (N), b = *p* < 0.05 vs. Hypertensive (H). Panel (**C**): AKT expression in left ventricle (*n* = 3–4 rats/group). a = *p* < 0.05 vs. Normotensive (N), b = *p* < 0.05 vs. Hypertensive (H), c = *p* < 0.05 vs. Hypertensive + Capsaicin (H + CS), d = *p* < 0.05 vs. Normotensive + Capsaicin (N + CS). z = Representative immunoblot. By Duncan’s post hoc test. Data are presented as means ± SEM.

**Figure 6 ijerph-16-03576-f006:**
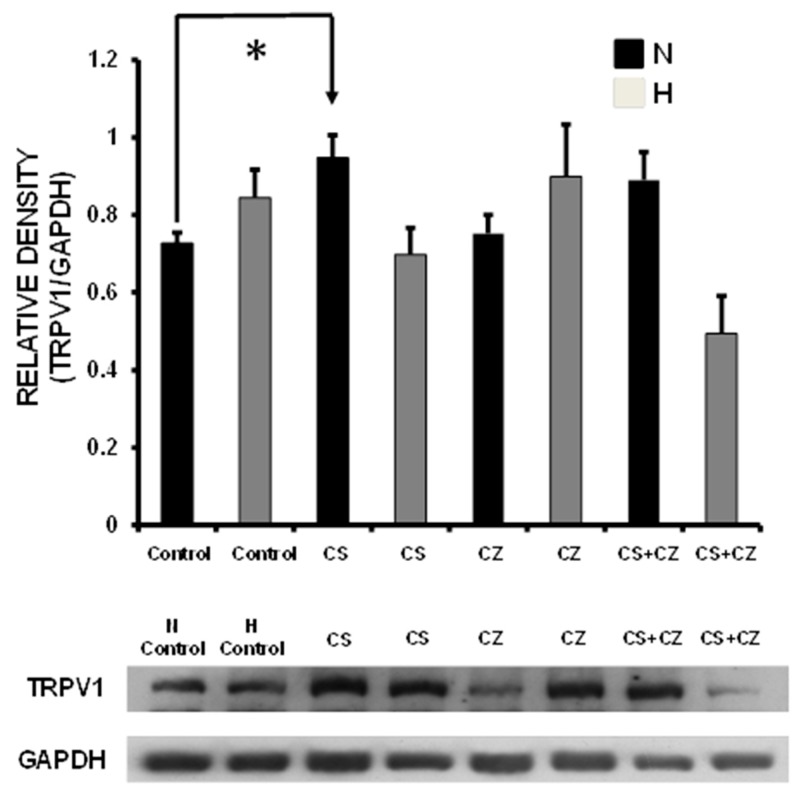
Expression of TRPV1 in the cardiac tissue from normotensive (N) and hypertensive (H) rats, in the absence and presence of capsaicin (CS), capsazepine (CZ), and capsazepine + capsaicin (CZ + CS). (*n* = 3–4 rats/group). By Duncan’s post hoc test. Data are presented as means ± SEM. (* *p* < 0.05 of N and H rats).

**Table 1 ijerph-16-03576-t001:** Effect of capsaicin and capsazepine in mean arterial pressure on normotensive and hypertensive rats.

Experimental Groups	Mean Arterial Pressure (mmHg)	
Treatment	Normotensive	Hypertensive
**Control**	118 ± 3	165 ± 4 *
**Capsaicin**	121 ± 2	147 ± 2 **
**Capsazepine**	130 ± 5	172 ± 3
**Capsazepine + Capsaicin**	125 ± 8	154 ± 9 ***

* Normotensive vs. Hypertensive; ** Hypertensive vs. Hypertensive + Capsaicin; *** Hypertensive vs. Hypertensive + Capsazepine + Capsaicin.

**Table 2 ijerph-16-03576-t002:** Effect of capsaicin and capsazepine on coronary vascular resistance in normotensive and hypertensive rats.

Experimental Groups	Coronary Vascular Resistance (mmHg/mL/min)	
Treatment	Normotensive	Hypertensive
**Control**	4 ± 0.3	11 ± 0.9 *
**Capsaicin**	5 ± 0.4	7 ± 0.4 **
**Capsazepine**	6 ± 0.4	9 ± 0.4
**Capsazepine + Capsaicin**	6 ± 0.5	6 ± 0.5 ***

* Normotensive vs. Hypertensive; ** Hypertensive vs. Hypertensive + Capsaicin; *** Hypertensive vs. Hypertensive + Capsazepine + Capsaicin.

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
