# Peer review of "The Role of the Activation of the TRPV1 Receptor and of Nitric Oxide in Changes in Endothelial and Cardiac Function and Biomarker Levels in Hypertensive Rats"

_ijerph, 2019, doi:10.3390/ijerph16193576_

Round 1
Reviewer 1 Report
In the paper by Torres-Narváez and co-workers, the effect of capsaicin on L-NAME treated rats, a model of hypertension, is characterized by pharmacological and biochemical approaches. They demonstrated that activation of the TRPV1 receptor and of the NO synthetic pathway reverts hypertension and biomarker levels in hypertensive rats.
The paper is interesting and results sound.
Comments:
It is necessary that figures 1, 2 and 4 should be implemented with statistical analysis as reported in the other figures and tables, using symbols or asterisks on the graph and explaining them in the figure legends. A mention to the results on blood pressure should be done in the abstract. The paper needs some editing and English check, as examples Line 28 …the levels of NO… Lines 131 e 136: please correct 0,22 um Line 214: … were obtained… Line 229: …control with β-actin… Line 331: … are increased… Line 381: FDE-3?? Line 417: sGS??Author Response
Response to Reviewer 1 Comments
Point 1: It is necessary that figures 1, 2 and 4 should be implemented with statistical analysis as reported in the other figures and tables, using symbols or asterisks on the graph and explaining them in the figure legends.
Response 1: The required statistical corrections to figures 1, 2 and 4 were done.
Point 2: A mention to the results on blood pressure should be done in the abstract.
Response 2: The values of blood arterial pressure are now indicated.
Point 3: The paper needs some editing and English check, as examples Line 28 …the levels of NO. Lines 131 to 136.
Response 3: Corrections to english language were done and the paper was checked again for possible mistakes.
Point 4: Please correct 0.22 um Line 214: … were obtained… Line 229: …control with β-actin… Line 331: … are increased… Line 381: FDE-3?? Line 417: sGS??
Response 4: The indicated corrections were done.
Reviewer 2 Report
In this study, the authors investigated the effects of TRPV1 activation on hypertensive rats relating to the bioavailability of nitric oxide. The issues raised in this study are as follows:
1. Abstract
: The first paragraph does not describe the purpose of the study.
: What does mean CS?
: What does mean H rats?
2. Introduction
: The purpose of the study is duplicated in the first and last paragraphs.
: A theoretical background on the association between NO synthesis and TRPV1 function in arterial hypertension must be provided.
: Theoretical background of the mechanism of action in arterial hypertension through activation or inhibition of TRPV1 must be described detail.
: The mechanism of action of capsaicin and capsazepine targeting TRPV1 must be described.
3. Methods
: How to make the capsaicin or capsazepine for S.C injection?
: Provide dosing schedule for capsaicin or capsazepine injection.
: Analysis of various factors、such as MDA, cGMP, and BH, etc., can be included within a large class method, "Capillary zone electrophoresis". The methodology needs to be refurbished.
4. Results
: For most bar graphs, no asterisks (*s) were shown, which is inconsistent with the description of Figure Legends.
: The results in Figure 1 should be analyzed by t-test.
: In Figure 2A, generation of NO in the hypertensive rat group should be comparable to or even more reduced in the capsazepine treated group as compared to the control group.
: In Figure 1D, MDA in the hypertensive rat group should be comparable to or even more increased in the capsazepine treated group as compared to the control group.
: In Table 2, the effect of coronary vascular resistance in the hypertensive group co-treated with capsaicin and capsazepine was similar to the treated with capsaisin alone. Was there no inhibitory effect of capsazepine?
: In Figure 4A and C, related factors in the hypertensive rat group should be comparable to or even more reduced in the capsazepine treated group as compared to the control group.
: In Figure 4B, BH2 in the hypertensive rat group should be comparable to or even more increased in the capsazepine treated group as compared to the control group.
: In Figure 5, all factors in the hypertensive group co-treated with capsaicin and capsazepine was more increased than the treated with capsaisin alone. Was there no inhibitory effect of capsazepine?
5. Discussion
: In hypertensive rats, with the reduction of NO, BH4, cGMP and AC decrease, whereas the expression of PDE-3 and MDA increases. It should be discussed what these results mean.
: The increase in NO is closely related to the increase in ROS. Concerns about the effects of ROS should be discussed.
As a whole, it has to go through a rigorous revision of grammatical errors and typos.
Author Response
Response to Reviewer 2 Comments
Abstract
Point 1: The first paragraph does not describe the purpose of the study.
Response 1: The purpose of the study is now stated in the abstract.
Point 2: What does mean CS?
Response 2: CS meaning (capsaisin) is now defined.
Point 3: What does mean H rats?
Response 3: The meaning of H (hypertensive) was added.
Introduction
Point 4: The purpose of the study is duplicated in the first and last paragraphs.
Response 4: The duplicity of the aim of the study was suppressed.
Point 5: A theoretical background on the association between NO synthesis and TRPV1 function in arterial hypertension must be provided.
Response 5: The theoreticas aspects related to the synthesis of NO and to the function of TRPV1 have been included as suggested.
Point 6: Theoretical background of the mechanism of action in arterial hypertension through activation or inhibition of TRPV1 must be described detail.
Response 6: Theoretical background relating to the mechanism of action of TRPV1 in hypertension have been added.
Point 7: The mechanism of action of capsaicin and capsazepine targeting TRPV1 must be described.
Response 7: The mechanism of action of capsaicin and capsazepine targetingTRPV1 are now stated.
Methods
Point 8: How to make the capsaicin or capsazepine for S.C injection?
Response 8: In lines 104, 106 and 110, we are describing how caspaicin and capsazepine were prepared and administered.
Point 9: Provide dosing schedule for capsaicin or capsazepine injection.
Response 9: The timing Schedule for capsaicin and capsazepine injection is now provided.
Point 10: Analysis of various factors、such as MDA, cGMP, and BH, etc., can be included within a large class method, "Capillary zone electrophoresis". The methodology needs to be refurbished.
Response 10: It is not posible to establish a general method of análisis for all of these molecules due to the different conditions needed for the separation of each molecule. The condition for each mlecule is independent. Therefore we summarized the methodologies employed mentioning only the detection parameters.
Results
Point 11: For most bar graphs, no asterisks (*s) were shown, which is inconsistent with the description of Figure Legends.
Response 11: The suggested corrections to figures were done.
Point 12: The results in Figure 1 should be analyzed by t-test.
Response 12: The Student`s T test was performed for figure 1.
Point 13: In Figure 2A, generation of NO in the hypertensive rat group should be comparable to or even more reduced in the capsazepine treated group as compared to the control group.
Response 13: The observation is correct; however, the physiopathological condition could explain the difference since there is a statistically significant difference between groups. We do not have, for the moment another explanation for this fact.
Point 14: In Figure 1D, MDA in the hypertensive rat group should be comparable to or even more increased in the capsazepine treated group as compared to the control group.
Response 14: The nature and chemical reactivity of capsazepine (polyphenol) do not allow for the molecule to act as an antioxidant agent for lipidic membranas; therefore, MDA levels (final product of lipoperoxidation) cannot or may not be increased.
Point 15: In Table 2, the effect of coronary vascular resistance in the hypertensive group co-treated with capsaicin and capsazepine was similar to the treated with capsaisin alone. Was there no inhibitory effect of capsazepine?
Response 15: This result may be due to the fact that the administration of the drugs was done for 4 days and therefore, the pharmacokinetics of distribution is at equilibrium. This may be different to what is reported in isolated organ models.
Point 16: In Figure 4A and C, related factors in the hypertensive rat group should be comparable to or even more reduced in the capsazepine treated group as compared to the control group.
Response 16: Polyphenols such as capsazepine have pleiotropic effects, which could explain other pharmacologic actions and not only the antagonistic effect on the receptor.
Point 17: In Figure 4B, BH2 in the hypertensive rat group should be comparable to or even more increased in the capsazepine treated group as compared to the control group.
Response 17: This result might be due to the administration of the drugs for 4 days. The pharmacokinetic distribution is in equilibrium and effects may be different from those found in isolated organ models. Furthermore, pleiotropic effects may be present.
Point 18: In Figure 5, all factors in the hypertensive group co-treated with capsaicin and capsazepine was more increased than the treated with capsaisin alone. Was there no inhibitory effect of capsazepine?
Response 18: The agonistic and antagonistic mechanisms are not the same. Both molecules (capsaicin and capsazepine, due to their polyphenolic nature stimulate the expression of enzymes. Capsazepine has an independent mechanisms that antagonizes the TRPV1 receptor. TRPV1.
Discussion
Point 19: In hypertensive rats, with the reduction of NO, BH4, cGMP and AC decrease, whereas the expression of PDE-3 and MDA increases. It should be discussed what these results mean.
Response 19: We have included a more detailed explanation in answer to the reviewer`s observation.
Point 20: The increase in NO is closely related to the increase in ROS. Concerns about the effects of ROS should be discussed.
Response 20: The requested information has been added.
Point 21: As a whole, it has to go through a rigorous revision of grammatical errors and typos.
Response 21: The paper was thoroughly checked for gramatical errors and typos.

Round 2
Reviewer 2 Report
There are still critical parts that the complement is incompleted.
Point 9: Provide dosing schedule for capsaicin or capsazepine injection.
Response 9: The timing Schedule for capsaicin and capsazepine injection is now provided
*Again, describe the method for administration and a dosing schedule during the study.
Point 10: Analysis of various factors、such as MDA, cGMP, and BH, etc., can be included within a large class method, "Capillary zone electrophoresis". The methodology needs to be refurbished.
Response 10: It is not posible to establish a general method of análisis for all of these molecules due to the different conditions needed for the separation of each molecule. The condition for each mlecule is independent. Therefore we summarized the methodologies employed mentioning only the detection parameters.
* The author did not understand the instruction. That means it can be included all in one chapter of "Capillary zone electrophoresis".
Point 11: For most bar graphs, no asterisks (*s) were shown, which is inconsistent with the description of Figure Legends.
Response 11: The suggested corrections to figures were done.
* The phrase on One-way ANOVA should also be deleted.
* The answers for Point 13-18 did not help with solving the issues at all. The author clearly amended the purpose of the study as follows: "The purpose of the present study was to analyze the actions of transient receptor potential vanilloid type 1 (TRPV1) agonist Capsaicin (CS) and of its antagonist Capsazepine (CZ), on cardiac function as well as endothelial biomarkers and some parameters related with nitric oxide (NO) release in L-NAME induced hypertensive rats"
If so, the role of antagonization of capsazepine on TRPV1 should be reflected in the results, but most of the results have not been reflected like anticipation. The ultimate purpose of the authors to use capsazepine is contradictory.
Author Response
Response to Reviewer 2 Comments (Round 2)
There are still critical parts that the complement is incompleted.
Point 9: Provide dosing schedule for capsaicin or capsazepine injection.
Response 9: The timing Schedule for capsaicin and capsazepine injection is now provided.
*Again, describe the method for administration and a dosing schedule during the study.
Response *: The method of administration was corrected. Schedules were indicated during the study in section 2.3. Experimental groups (lines 101-108).
Point 10: Analysis of various factors such as MDA, cGMP, and BH, etc., can be included within a large class method, "Capillary zone electrophoresis". The methodology needs to be refurbished.
Response 10: It is not possible to establish a general method of analysis for all of these molecules due to the different conditions needed for the separation of each molecule. The condition for each molecule is independent. Therefore we summarized the methodologies employed mentioning only the detection parameters.
* The author did not understand the instruction. That means it can be included all in one chapter of "Capillary zone electrophoresis".
Response *: The capillary zone electrophoresis methodologies for biopterins (section 2.7.), MDA (section 2.8.), cGMP (section 2.9.) and PDE-3 (section 2.10.) were removed and added in a section of capillary zone electrophoresis (line 155, comment LdV29).
Point 11: For most bar graphs, no asterisks (*s) were shown, which is inconsistent with the description of Figure Legends.
Response 11: The suggested corrections to figures were done.
* The phrase on One-way ANOVA should also be deleted.
Response *: The phrase “Two-way ANOVA followed” was deleted from the text (lines 245 and 299) and from the legend captions (lines 266, 288 and 360).
* The answers for Point 13-18 did not help with solving the issues at all. The author clearly amended the purpose of the study as follows: "The purpose of the present study was to analyze the actions of transient receptor potential vanilloid type 1 (TRPV1) agonist Capsaicin (CS) and of its antagonist Capsazepine (CZ), on cardiac function as well as endothelial biomarkers and some parameters related with nitric oxide (NO) release in L-NAME induced hypertensive rats" If so, the role of antagonization of capsazepine on TRPV1 should be reflected in the results, but most of the results have not been.
Response *: Justifications were added explaining such behavior (line 389, comment LdV84) and (line 458, comment LdV92). References 52 (line 609, comment LdV95), 53 (line 610, comment LdV96) and 54 (line 611, Comment LdV97) were added.

Round 3
Reviewer 2 Report
The authors newly made the following assertion through Line-449-450:"However, the effect of the competitive antagonist CZ on the TRPV1 receptor, is only evident in normotensive animals without it being present in the hypertensive ones (Figure 2. A.)." If so, no effect of capsazepine should be induced in hypertensive rats. However, 1. Figure 2A-C
In the H group, treatment with capsaicin significantly increases the NO level as compared to the control. The level of NO was not increased in the capsaicin and capsazepine combination group as much as the capsaicin alone group, but rather similar to capsazepine single group.
2. Figure 3A and B
In the H group, treatment with capsaicin significantly increases the cardiac mechanical performance (CMP) and NO level as compared to the control. The level of CMP and NO was not increased in the capsaicin and capsazepine combination group as much as the capsaicin alone group, but rather similar to capsazepine single group. It was similar pattern with the N group.
3. Table 2
In the Normotensive group, there was no effect of capsazepine. But the results of the hypertensive group are acceptable if there is any effect of capsazepine in hypertensive rats.
4. Figure 4A-C
If there was no effect of the competitive antagonist CZ on the TRPV1 receptor in hypertensive rats, the capsazepine single group should have similar trends as the control group. However, it was significantly increased or decreased as compared to the control group.
5. Figure 5C
In the H group, treatment with capsazepine significantly decreases the AKT level as compared to the control group.
6. Figure 6
Depending on the result, it has been proven that the effect of capsazepine only works in normotensive rats. Rather, TRPV1 expression was reduced in the hypertensive rats treated with capsaicin as compared to the control group. Summarizing all these results, the effects of capsaicin and capsazepine in hypertensive rats are inconsistent with the author's assertion. The author should be able to cope with this logical contradiction